# Meal Timing and Sleep Health Among Midlife Mexican Women During the Early Stages of the COVID-19 Pandemic

**DOI:** 10.3390/nu16223967

**Published:** 2024-11-20

**Authors:** Maymona Al-Hinai, Afnan Mohy, Martha María Téllez-Rojo, Libni A. Torres-Olascoaga, Luis F. Bautista-Arredondo, Alejandra Cantoral, Karen E. Peterson, Erica C. Jansen

**Affiliations:** 1Department of Food Science and Human Nutrition, College of Agriculture and Marine Science, Sultan Qaboos University, Muscat 123, Oman; maymona@squ.edu.om; 2Department of Nutritional Sciences, University of Michigan School of Public Health, Ann Arbor, MI 48109, USA; afnanmoh@umich.edu (A.M.); karenep@umich.edu (K.E.P.); 3Center for Nutrition and Health Research, National Institute of Public Health, Cuernavaca 62100, Mexico; mmtellez@insp.mx (M.M.T.-R.); libniavib@gmail.com (L.A.T.-O.); lbautista@insp.mx (L.F.B.-A.); 4Department of Health, Iberoamericana University, Mexico City 01219, Mexico; alejandra.cantoral@insp.mx

**Keywords:** meal habits, meal timing, sleep duration, sleep latency, sleep quality

## Abstract

Background/Objectives: This study aimed to examine associations between meal timing habits and sleep health in midlife Mexican women. Methods: Data comprised 379 midlife Mexican women who participated in a phone survey conducted within the Early Life Exposures in Mexico to Environmental Toxicants (ELEMENT) project during the early stages of the COVID-19 pandemic. Women answered questions related to meal habits and sleep duration, latency, and quality. We used linear regression to investigate the associations between meal timing, frequency of meals/snacks, eating window (duration between first and last eating occasion of the day), duration between last meal, bedtime, sleep duration, and logistic regression to examine the associations between meal timing, sleep latency, and sleep quality, adjusting for confounders. Results: Later timing of meals throughout the day, and a shorter interval between the last meal of the day and bedtime, were associated with prolonged sleep latency and worse sleep quality. Associations with sleep duration were mixed: a longer eating window and a later largest and last meal were each associated with shorter sleep duration, while a later first meal and a shorter interval between the last meal of the day and bedtime were associated with longer sleep duration. Conclusions: Meal timing habits are associated with sleep duration, latency, and quality in midlife women.

## 1. Introduction

Sleep is becoming more widely acknowledged as a significant factor in maintaining good health [1]. A number of chronic health conditions, including obesity, type 2 diabetes, high blood pressure, and heart disease, have been linked to short sleep duration and poor sleep quality [2,3]. The American Heart Association’s “Life’s Essential 8”, which identifies critical measures for improving cardiovascular health, now includes obtaining enough sleep, defined as between 7–9 h for adults, in order to promote healing, improve cognitive function, and lower the risk of chronic diseases [4,5]. Besides the amount of time spent sleeping, the timing of sleep also has an effect on health. For example, people with later chronotypes, meaning that they have a preference for later bedtimes and wake times, have a higher risk of health problems, such as metabolic dysfunction, cardiovascular disease (CVD), and mental health concerns as well as a higher risk of both morbidity and mortality due to these causes [6,7,8,9]. Long sleep latency, or the amount of time it takes to fall asleep, is a marker of overall lower sleep quality and potential underlying sleep disorder (i.e., insomnia) [10,11]. Long sleep latency and insomnia may also have independent effects on cardiometabolic and mental health [12,13].

Numerous studies now underscore the importance of higher diet quality as a modifiable factor to promote multiple aspects of adequate sleep quality [14]. However, as highlighted in a recent scoping review, a significant gap exists in the literature related to various aspects of meal timing [15]. Temporal variations in meal timing, including the clock time (and circadian time) of individual meals, the length of the eating window (hours from first to last eating occasion), and daily fluctuations in the eating window or individual meals, are now being recognized to have a substantial impact on health, particularly metabolic health [16,17,18,19]. Some studies hint at the importance of examining eating timing in relation to sleep. For instance, a study that aimed to explore the relationship between sleep timing, dietary patterns, and BMI in 52 US adults (48% female) found that individuals classified as “late sleepers” tended to have shorter sleep duration, later mealtimes, and higher caloric intake, particularly after 8:00 PM, compared to “normal sleepers” [18]. Another study among US adults (53% female) was conducted to investigate the correlation between mealtime and sleep duration, with a specific focus on the effects of consuming food or beverages within one hour before going to bed [20]. They indicated that consuming food shortly before going to bed may potentially extend the duration of sleep, but also negatively impact the quality of sleep, resulting in more frequent awakenings during the night. Therefore, choosing a longer time period between meals and going to bed appeared to be linked to improved sleep outcomes. 

One important limitation of the prior literature is the lack of examination of meal timing and sleep specifically of midlife women. An important role of eating timing in midlife women is plausible since changes in circadian rhythms occur during perimenopause, with postmenopausal women showing lower daily rhythmicity in sleep/wake behaviors than premenopausal women [21]. Perimenopause is also a time when many women experience sleep disruptions (e.g., insomnia and nighttime awakenings) due to the menopausal transition [22]. Although some studies have noted associations between healthier diets overall on sleep quality in midlife women [23], only one examined eating timing behaviors, finding that evening food consumption was associated with shorter self-reported sleep duration [15,24]. Given the scarcity of research on the timing of eating in relation to sleep among midlife women, the primary goal of this study was to investigate the association between eating timing and sleep health in a population of midlife women from Mexico City, Mexico. This study was completed during the early stages of the COVID-19 pandemic, a period of time when sleep and dietary patterns changed in many populations across the globe [25]. In midlife women specifically, declines in sleep quality were noted whereas changes in diet quality were mixed [26,27,28,29]. Although the goal of the present study was not to evaluate the effect of the pandemic on eating and sleep in this population, it is important to acknowledge the specific context during which the meal timing and sleep quality in this Mexico City population of midlife women were assessed.

## 2. Materials and Methods

### 2.1. Study Population

This study includes participants enrolled in a phone survey conducted during the COVID-19 pandemic within the Early Life Exposures in Mexico to Environmental Toxicants (ELEMENT) project. The study design and data collection procedure of the larger cohort study has been published previously [30]. Briefly, between 1997–2004, 1012 mother–child dyads were recruited from maternity hospitals serving low- to middle-income populations in Mexico City. Between June 2020 and June 2021, during the COVID-19 pandemic, a subsample of 595 women at midlife were re-enrolled for a follow-up study. A telephone interview was conducted to collect data about demographic characteristics, socioeconomic status, occupation, sleep, meal habits, food security, and health. The analytic sample included 379 women with complete data on sleep and meal habits.

### 2.2. Meal Habits

Survey questions were used to determine meal habits, including the usual timing and frequency of meals. Participants were asked to report the number of meals and snacks in a typical day and the time of consuming the first, middle, and last meal as well as the time of the largest meal of the day (which may or may not coincide with the first, middle, or last). We categorized the number of meals/snacks per day into 1 or 2 meals/snacks, 3 meals/snacks, and 4 or more meals/snacks. The eating window was calculated as the number of hours between consumption of the first and last meal or snack. In addition, we estimated the duration in hours between the last meal/snack and bedtime.

### 2.3. Sleep Outcomes

Sleep data were self-reported using select questions from the Pittsburgh Sleep Quality Index (PSQI), a validated questionnaire that subjectively measures various aspects of sleep including sleep duration, sleep latency, and sleep quality [31]. Sleep duration was assessed on weekdays and on the weekends by asking “Over the past month: what has usually been your bedtime?” and “What time have you usually gotten up in the morning during the last month?”. Sleep latency was assessed by asking “How long has it taken to fall asleep, normally, the nights of the last month?” on weekdays and during the weekend. We measured sleep duration by subtracting the time taken to fall asleep (sleep latency) from the total reported hours of sleep. Sleep latency was dichotomized to normal (≤30 min/night) or prolonged (>30 min/night), in accordance with prior literature [32,33]. Participants were asked to rate sleep quality in the last month and the overall sleep quality (longer-term), with four options: “very bad”, “fairly bad”, “fairly good”, or “very good”. We categorized sleep quality as poor if the response was reported as “fairly bad” or “very bad” and as good if the response was “fairly good” or “very good”.

### 2.4. Covariates

Covariate selection was based on a prior knowledge and included: age, education level (less than high school, high school, more than high school), socioeconomic status ((estimated using wealth index based on measures of housing (constructing materials used for flooring), number of rooms, type of toilet, electricity, and ownership of assets (water-heater, TV, computer, radio, stove, washing machine, microwave, and vacuum cleaner) and categorized into tertiles)), marital status (married, cohabit, single/divorced/separated), alcohol consumption (never drinker, former drinker, current drinker), smoking (never smoker, former smoker, current smoker), physical activity (the sum metabolic equivalents from vigorous and moderate activities obtained from the International Physical Activity Questionnaire [34] and categorized into “no activity” (0 metabolic equivalents) or “any activity”(>0)), working status (working/student, retired, not working), diagnosis of a mental health condition (yes, no), and food insecurity (assessed using the Latin-American Scale of Food Security (ELCSA) that measures household food access and utilization and categorized into no food security, mild, moderate, severe) [35].

### 2.5. Statistical Analysis

We first conducted descriptive analyses of the main characteristics of the study sample reporting mean (standard deviation) for continuous variables and proportions for categorical variables. Next, we examined the distribution of study exposures (eating window, number of meals or snacks/day, time of first meal, time of mid-day meal, time of last meal, and time of the largest meal) across sociodemographic and lifestyle characteristics. We tested the associations with non-parametric Mann–Whitney U tests for variables with two categories and Kruskal–Wallis tests for variables with more than two categories. Linear regression was used to examine the associations between study exposures and continuous sleep duration. Logistic regression was used to examine the associations between study exposures and the binary outcomes of prolonged sleep latency and poor sleep quality (in separate models). For each sleep outcome, 0 was the more optimal sleep category (sleep latency ≤ 30 min or good sleep quality, rated as “fairly good” or “very good), and 1 was the less optimal (>30 min sleep latency or poor sleep quality, rated as “very bad” or “fairly bad”). In the multivariable models, we adjusted for age, education level, socioeconomic status, marital status, alcohol consumption, smoking, physical activity, working status, mental health condition, and food insecurity. We conducted several sensitivity analyses including removing subjects with zero-time duration between last meal and bedtime, examining sleep duration without subtracting sleep latency, and examining sleep latency in the continuous scale. Diagnostic checking of model assumptions was conducted to check for any violation that could affect the results. All statistical analyses were performed using SAS 9.4 (SAS Institute Inc., Cary, NC, USA).

## 3. Results

In this analytic sample of 379 women, aged 46.5 ± 5.8 years on average, eating patterns were examined. The average time for the first meal was 9:13, with the mid-day meal and the last meal typically occurring around 15:18 and 20:53, respectively (Table 1). The mean time for the largest meal was 13:49. On average, participants had an eating window of 11.7 ± 1.7 h, with approximately 3.6 ± 0.8 meals or snacks and 2.5 ± 1.3 h between the last meal and bedtime. Participants reported an average weekday sleep duration of 7.9 ± 1.6 h and a weekend sleep duration of 8.6 ± 1.7 h. About a quarter of the participants reported prolonged sleep latency, 15% reported poor sleep quality overall, and 29% reported poor sleep quality in the last month.

Table 2 shows the distribution of study exposures across demographics and lifestyle characteristics. Higher socioeconomic status, being physically active, no alcohol consumption, and no food insecurity were each associated with a higher number of meals and snacks consumed per day. Women who worked consumed their first meal earlier and had a shorter eating window compared to those who did not work or were retired. In contrast, women who smoked had later mealtimes compared to non-smokers.

The multivariable associations between meal habits and sleep outcomes were investigated, adjusting for age, education, marital status, socioeconomic status (SES), smoking, alcohol consumption, physical activity, mental health issues, and food insecurity (Table 3). A higher number of meals or snacks exhibited a significant inverse association with poor sleep quality in the last month (OR = 0.741, 95% CI [0.553, 0.993]); however, adjusting for confounders attenuated the association. The timing of the first meal was associated with weekday sleep duration and sleep latency. Specifically, consuming the first meal at a later time was associated with longer sleep duration (β = 0.247, 95% CI [0.135, 0.360]) but also with prolonged sleep latency (OR = 1.243, 95% CI [1.041, 1.508]). On the other hand, later timing of the mid-day meal showed significant inverse associations with sleep duration on weekends (β = −0.177, 95% CI [−0.305, −0.048]). Additionally, the time of the last meal exhibited negative associations with sleep duration on both weekdays (β = −0.185, 95% CI [−0.302, −0.068]) and weekends (β = −0.259, 95% CI [−0.391, −0.128]), and positive associations with overall poor sleep quality (OR = 1.337, 95% CI [1.048, 1.706]) and sleep quality in the last month (OR = 1.388, 95% CI [1.141, 1.690]). Finally, a later time for the largest meal was linked to shorter sleep duration on weekends (β = −0.069, 95% CI [−0.130, −0.007]), as well as prolonged sleep latency in both weekdays (OR = 1.126, 95% CI [1.028, 1.233]) and weekends (OR = 1.116, 95% CI [1.019, 1.223]).

Overall, a longer eating window was inversely associated with sleep duration on weekdays (β = −0.292, 95% CI [−0.382, −0.201]). A longer duration from the last meal to bedtime was associated with shorter sleep duration on weekdays (β = −0.176, 95% CI [−0.293, −0.059]) and weekends (β = −0.283, 95% CI [−0.414, −0.153]), but lower odds of long sleep latency weekdays (OR = 0.738, 95% CI [0.601, 0.906]) and weekends (OR = 0.765, 95% CI [0.626, 0.937]), and better overall sleep quality (OR = 0.761, 95% CI [0.591, 0.980]). Notably, results from sensitivity analyses closely aligned with the initial findings.

## 4. Discussion

In this study of 379 middle-aged women, we examined the cross-sectional associations between meal habits and sleep duration, latency, and quality. We uncovered several significant associations between meal timing and sleep outcomes. Later timing of the first meal was associated with longer sleep duration but prolonged sleep latency during the weekdays, whereas mid-day meal timing was associated with shorter sleep duration on the weekends. Moreover, later timing of eating the last meal was associated with shorter sleep duration in both weekdays and weekends and with poorer sleep quality. The time of the largest meal, which is usually in the afternoon, was associated with sleep duration and prolonged sleep latency. Regarding the duration of eating, a longer eating window was associated with shorter sleep duration during the weekdays, whereas a longer fasting duration between the last meal and bedtime was associated with shorter sleep duration but with better sleep quality and shorter time to fall asleep.

The finding that later timing of eating, including later timing of the first meal, midday meal, the closest meal/snack to bedtime, and the largest meal (not necessarily mutually exclusive) was related to poorer sleep health and is consistent with some previous studies [19,20,24,36,37]. Notably, a small study of 52 US young adults found that later eating patterns were associated with later bedtimes, wake times, and lower sleep efficiency based on 7-day actigraphy [36]. Another study of Brazilian adults aged 19–45 showed that higher nocturnal calorie intake correlated with longer sleep latency and lower sleep efficiency in women but not men [37]. In addition, a study of 1098 midlife Finnish women found that those reporting the highest consumption of food in the evening were more likely to have self-reported short sleep duration, although this association did not hold after accounting for multiple comparisons [24]. A few other studies examined the specific duration of time between the last meal and bedtime in relation to sleep. Nogueira et al. (2021) found that a shorter interval between the last meal and sleep onset led to an increase in diurnal sleep duration [19]. Similarly, analysis of the American Time Use Survey showed that diary-reported eating or drinking less than one hour before bedtime was associated with longer sleep duration but increased wake after sleep onset (WASO). In particular, women and men who consumed food or drinks within an hour of bedtime experienced extended sleep duration by 35 and 25 min, respectively, but also they had higher odds of experiencing WASO [20]. The present findings similarly indicated that a shorter duration between the last meal and bedtime related to longer sleep duration but potentially worse overall sleep quality (there was a marginally significant association with sleep quality). Nonetheless, not all studies have reported associations between late-night eating and sleep [38]. Furthermore, a recent scoping review on chrono-nutrition and sleep highlights inconsistencies in the literature on late eating and sleep quality, with experimental studies examining effects on objectively measured sleep quality showing contrasting results [15].

There are a few potential mechanisms to explain an association between later eating timing and worse sleep. The first possible explanation is related to circadian effects. Eating is a driver of circadian rhythms, especially those found in peripheral organs [39]. Although sleep is primarily controlled via the central clock, misalignment between the peripheral and central circadian rhythms may still impact sleep [40]. Additionally, digestive processes from food consumed too close to sleep onset could make it difficult to fall asleep or cause increased awakenings after sleep onset [41]. Spicy or fatty foods especially could trigger temperature-related sleep disturbances [42] and/or insomnia [43]. Other studies suggest that both higher consumption of fat and carbohydrates before bedtime are associated with a longer diurnal sleep onset latency [19]. Although we did not examine the individual foods consumed; late-night foods are more often highly processed, with more refined carbohydrates and saturated fats than foods consumed earlier in the day [44,45].

Few studies have examined overall eating duration windows in relation to sleep quality. A study of 296 Brazilian adults with obstructive sleep apnea showed that an eating duration > 12 h was related to shorter self-reported sleep duration than eating for ≤12 h per day [46] whereas a study of 52 US young adults reported no association between eating duration and actigraphy-assessed sleep parameters [36]. Studies examining the frequency of meals/snacks in relation to sleep quality are also scarce. Interestingly, in our sample, a higher number of meals/snacks throughout the day was associated with better sleep quality. As a possible explanation, a higher number of meals/snacks could reflect less meal skipping, particularly breakfast, which can be associated with higher consumption of calories in subsequent meals [47], lower overall diet quality, and/or higher consumption of fat and carbohydrates in the evening [48]. Indeed, previous research that was completed among university students showed that skipping breakfast, consuming late-night snacks, and replacing meals with snacks are associated with poorer overall sleep quality further emphasizes the critical role of regular meal timing and healthy eating habits in maintaining good sleep quality [49]. However, a review on meal timing and sleeping energy metabolism found that while time-restricted eating such as skipping breakfast or dinner affected the time course of energy metabolism, it did not negatively affect the subjective quality of sleep [50].

Potential mechanisms to explain associations with the duration of eating are likely related to the circadian misalignment mechanisms mentioned above. For example, skipping meals earlier in the day may lead to overconsumption later in the day, including late-night snacking [47]. Long eating windows can also result in circadian misalignment if eating occurs during a time that should be a period of fasting based on external cues (i.e., light).

Given the bidirectional nature of eating and sleep relationships, reverse causation may also help explain the associations we observed. For example, short sleepers may start eating earlier in the day due to longer awake times. A survey study from Sweden during the COVID-19 pandemic revealed a delay in sleep timing that was accompanied by a corresponding shift in the timing of early meals, but not with late meals [51]. The present study also occurred during the pandemic, which is another important contextual factor to highlight. Both eating and sleep times may have differed in comparison to pre-pandemic life, with the implication that associations we observed here may be either stronger or weaker than associations observed in a study conducted pre- or post-pandemic. As an example, in unadjusted analysis, there was an association between fewer meals and poorer sleep quality (Table 3). However, after we adjusted for food insecurity, which was much higher in our sample specifically during the pandemic [52], and which could also impede sleep [53], the association became null. Another example of a potential role of the pandemic on results was the fact that the association between time from last meal to bedtime with sleep quality was only apparent for “overall sleep quality” (more representative of usual sleep quality) and not “sleep quality in last month” (more subject to circumstances related to the pandemic). Future work is needed to evaluate these associations in the post-pandemic era, ideally with better markers, i.e., actigraphy-assessed sleep.

One of the key strengths of the study is the specific focus on midlife women. Our findings on eating timing are complementary to a growing body of evidence that later eating timing and less consistent meal timing are related to worse cardiometabolic profiles among midlife women [54]. Taken together, these findings suggest the need for future research on the intersections of eating timing, sleep, and cardiometabolic health among women in midlife. Nonetheless, there are limitations to highlight as well. The cross-sectional design of this study limits our ability to ascertain causal inferences. Moreover, information was collected via phone survey, which can be subject to recall bias. Related to the survey format of the COVID-19 study, we did not have access to objective sleep assessments via actigraphy in order to validate self-reported sleep duration. While we examined the timing of meals, we did not measure the content and the amount of food eaten, which may affect sleep. Nor did we try to differentiate between meals and snacks, which can be an arbitrary distinction. Yet, one study found that relying on snacks as compared to meals was associated with greater daytime sleepiness [24], one proxy for poor sleep quality. Due to the abbreviated nature of the phone survey, we were not able to assess all potential confounders, including menopausal status, body mass index, and other eating behaviors. One eating behavior that may be relevant to meal timing is self-regulation. Specifically, women may be manipulating their diet for health reasons [55], including by altering meal timing. However, self-regulation of eating may not be independently related to sleep outcomes and therefore would cause residual confounding. Finally, the short survey format also did not allow us to fully evaluate changes in sleep due to the pandemic, although we did ask about sleep quality in the previous month compared to longer-term sleep health.

## 5. Conclusions

In summary, our findings suggest that meal timing and habits may play a significant role in sleep duration, latency, and quality among middle-aged women. Future longitudinal studies are needed to establish causality and further explore the mechanisms underlying this association.

## Figures and Tables

**Table 1 nutrients-16-03967-t001:** Characteristics of the study population, *n* = 379.

Variable	*n* or Mean	% or SD
Age (years)		
33–42	116	30.61
43–52	206	54.35
53–63	57	15.04
Education		
<High school	190	50.13
High school	134	35.36
>High school	55	14.51
Marital status		
Married	269	70.98
Cohabiting	66	17.41
Single, separated, divorced	44	11.61
Socioeconomic status		
Tertile 1	269	70.98
Tertile 2	66	17.41
Tertile 3	44	11.61
Work in the last week		
Yes	237	62.53
No	135	35.62
Retired	7	1.85
Smoking		
Never	151	39.84
Former	143	37.73
Current	85	22.43
Alcohol consumption		
Never	49	12.93
Former	43	11.35
Current	287	75.73
Physical activity		
None reported	253	66.75
Any reported	126	33.25
Mental health issues		
Yes	49	12.93
No	330	87.07
Food insecurity		
None	169	44.59
Mild	143	37.73
Moderate	39	10.29
Severe	28	7.39
Sleep duration weekday (hours)	7.88	1.56
Sleep duration weekend (hours)	8.57	1.74
Prolong sleep latency weekday	94	24.80
Prolong sleep latency weekend	93	24.54
Poor sleep quality overall	56	14.78
Poor sleep quality last month	109	28.76
Number of meals/snacks	3.64	0.84
Time of first meal	9:13:10	1:24:04
Time of mid-day meal	15:18:47	1:23:19
Time of last meal	20:52:02	1:20:35
Time of largest meal	13:49:50	2:56:20
Eating window (hours from 1st to last eating occasion)	11.64	1.69
Time from last meal to bedtime (hours)	2.30	1.34

**Table 2 nutrients-16-03967-t002:** Meal timing characteristics according to categories of demographics and lifestyle characteristics in midlife women from Mexico City.

Variable	Number of Meals/Snacks	Time of First Meal	Time of Mid-Day Meal	Time of Last Meal	Time of Largest Meal	Eating Window	Duration from Last Meal to Bedtime
Age (years)							
33–42	3.7 (0.9)	9.3 (1.4)	15.3 (1.3)	20.8 (1.3)	14.2 (3.0)	11.5 (1.7)	2.5 (1.3)
43–52	3.6 (0.8)	9.1 (1.4)	15.2 (1.5)	20.9 (1.4)	13.6 (2.9)	11.7 (1.8)	2.5 (1.4)
53–63	3.5 (0.7)	9.3 (1.4)	15.6 (1.1)	21.0 (1.2)	13.8 (2.7)	11.8 (1.3)	2.4 (1.3)
*p*-value *	0.5535	0.5227	0.5147	0.6567	0.1946	0.5400	0.7390
Education							
<High school	3.6 (0.8)	9.2 (1.5)	15.3 (1.4)	20.9 (1.4)	14.1 (2.9)	11.7 (1.8)	2.5 (1.4)
High school	3.7 (0.9)	9.3 (1.4)	15.2 (1.4)	20.9 (1.4)	13.6 (3.0)	11.6 (1.6)	2.5 (1.3)
>High school	3.7 (0.8)	9.3 (1.1)	15.4 (1.1)	20.9 (1.1)	13.5 (2.9)	11.6 (1.3)	2.6 (1.4)
*p*-value	0.2360	0.7755	0.5593	0.8867	0.3188	0.7810	0.8568
Marital status							
Married	3.6 (0.8)	9.2 (1.4)	15.4 (1.3)	20.9 (1.3)	13.6 (3.1)	11.7 (1.6)	2.5 (1.3)
Cohabit	3.7 (0.8)	9.3 (1.4)	15.2 (1.6)	21.0 (1.5)	14.6 (2.1)	11.6 (1.9)	2.3 (1.3)
Single, separated, divorced	3.8 (0.8)	9.1 (1.2)	15.0 (1.4)	20.5 (1.4)	14.3 (2.6)	11.3 (1.6)	2.9 (1.5)
*p*-value	0.3852	0.5316	0.3304	0.1391	0.0991	0.7534	0.0963
Socioeconomic status							
Tertile 1	3.6 (0.8)	9.2 (1.6)	15.4 (1.5)	21.0 (1.3)	14.2 (3.1)	11.8 (1.8)	2.5 (1.3)
Tertile 2	3.5 (0.8)	9.3 (1.4)	15.1 (1.5)	20.8 (1.5)	13.5 (2.9)	11.5 (1.8)	2.6 (1.4)
Tertile 3	3.8 (0.9)	9.2 (1.2)	15.5 (1.2)	20.9 (1.2)	13.8 (2.9)	11.7 (1.5)	2.5 (1.2)
*p*-value	**0.0309**	0.5959	0.0879	0.5665	0.1975	0.2641	0.9392
Work last week							
Yes	3.6 (0.8)	9.0 (1.4)	15.3 (1.4)	20.9 (1.3)	14.0 (3.0)	11.6 (1.6)	2.5 (1.4)
No	3.6 (0.8)	9.6 (1.3)	15.4 (1.4)	20.9 (1.4)	13.6 (2.8)	11.6 (1.6)	2.5 (1.3)
Retired	3.7 (0.8)	8.6 (1.7)	15.7 (0.8)	20.9 (0.9)	13.5 (2.9)	12.0 (1.8)	2.1 (1.6)
*p*-value	0.5218	**0.0003**	0.2519	0.9123	0.8967	**0.0004**	0.4579
Smoking							
Never	3.6 (0.7)	9.2 (1.3)	15.2 (1.1)	20.8 (1.3)	13.5 (3.1)	11.6 (1.6)	2.4 (1.2)
Former	3.7 (0.8)	9.3 (1.5)	15.3 (1.6)	20.8 (1.4)	13.9 (2.7)	11.5 (1.7)	2.6 (1.5)
Current	3.7 (0.9)	9.2 (1.5)	15.4 (1.5)	21.3 (1.3)	14.4 (3.0)	12.1 (1.8)	2.4 (1.2)
*p*-value	0.6337	0.8530	0.2713	**0.0086**	0.0637	0.0925	0.5596
Alcohol consumption							
Never	3.7 (0.8)	8.9 (1.4)	15.3 (1.0)	20.9 (1.1)	13.3 (3.1)	11.9 (1.5)	2.2 (1.0)
Former	3.3 (0.6)	9.4 (1.6)	15.3 (1.3)	20.9 (1.3)	14.1 (2.6)	11.5 (1.4)	2.6 (1.3)
Current	3.6 (0.8)	9.2 (1.4)	15.3 (1.5)	20.9 (1.4)	13.9 (3.0)	11.6 (1.8)	2.5 (1.4)
*p*-value	**0.0207**	0.2161	0.6822	0.9867	0.4546	0.3429	0.3774
Physical activity							
None reported	3.5 (0.8)	9.3 (1.4)	15.3 (1.4)	20.9 (1.4)	13.9 (3.0)	11.6 (1.7)	2.5 (1.4)
Any reported	3.7 (0.9)	9.1 (1.4)	15.3 (1.4)	20.9 (1.2)	13.8 (2.9)	11.8 (1.7)	2.5 (1.2)
*p*-value	**0.0187**	0.2774	0.8810	0.7376	0.8989	0.1705	0.9864
Mental health issues							
Yes	3.6 (0.8)	9.2 (1.4)	15.4 (1.6)	20.8 (1.4)	13.2 (3.0)	11.6 (1.8)	2.8 (1.5)
no	3.6 (0.8)	9.2 (1.4)	15.3 (1.4)	20.9 (1.3)	13.9 (2.9)	11.7 (1.7)	2.5 (1.3)
*p*-value	0.4594	0.7278	0.3309	0.7085	0.1189	0.5153	0.2333
Food insecurity							
No insecurity	3.8 (0.9)	9.2 (1.3)	15.2 (1.3)	20.8 (1.2)	13.6 (2.7)	11.6 (1.5)	2.5 (1.3)
Mild	3.6 (0.8)	9.2 (1.4)	15.2 (1.4)	20.8 (1.5)	14.2 (3.1)	11.6 (1.9)	2.6 (1.4)
Moderate	3.5 (0.8)	9.2 (1.6)	15.6 (1.6)	21.1 (1.5)	13.1 (3.6)	11.9 (1.9)	2.3 (1.4)
Severe	3.3 (0.5)	9.5 (1.4)	15.8 (1.3)	21.2 (1.1)	14.3 (2.6)	11.7 (1.4)	2.3 (1.2)
*p*-value	**0.0084**	0.7935	0.1608	0.4473	0.1822	0.8102	0.7217

* *p*-values shown throughout the table from Mann–Whitney U tests for variables with two categories and Kruskal–Wallis tests for variables with more than two categories. Bolded text indicates *p* < 0.05.

**Table 3 nutrients-16-03967-t003:** Multivariable associations between meal habits and sleep characteristics in midlife women from Mexico City ^1^.

	Sleep Duration (Weekday)	Sleep Duration (Weekend)	Prolonged Sleep Latency ^2^ (Weekday)	Prolonged Sleep Latency ^2^ (Weekend)	Overall Sleep Quality ^3^	Sleep Quality Last Month ^3^
	β (95% CI)	β (95% CI)	OR (95% CI)	OR (95% CI)	OR (95% CI)	OR (95% CI)
Number of meals/snacks						
Crude	−0.051 (−0.244, 0.142)	0.033 (−0.183, 0.248)	0.994 (0.747, 1.322)	0.883 (0.658, 1.186)	0.886 (0.618, 1.271)	0.741 (0.553, 0.993)
Adjusted	−0.056 (−0.255, 0.142)	0.018 (−0.207, 0.242)	1.093 (0.799, 1.494)	0.926 (0.671, 1.278)	0.978 (0.661, 1.446)	0.770 (0.561, 1.057)
Time of first meal						
Crude	0.273 (0.163, 0.382) *	−0.082 (−0.208, 0.043)	1.291 (1.084, 1.537) *	1.235 (1.039, 1.469) *	1.060 (0.865, 1.300)	1.183 (1.005, 1.392) *
Adjusted	0.247 (0.135, 0.360) *	−0.117 (−0.247, 0.013)	1.253 (1.041, 1.508) *	1.205 (0.998, 1.456)	0.997 (0.787, 1.263)	1.158 (0.966, 1.388)
Time of mid-day meal						
Crude	0.017 (−0.096, 0.131)	−0.185 (−0.311, −0.059) *	1.200 (1.004, 1.434) *	1.041 (0.878, 1.234)	1.112 (0.900, 1.374)	1.139 (0.965, 1.345)
Adjusted	0.042 (−0.073, 0.156)	−0.177 (−0.305, −0.048) *	1.162 (0.966, 1.398)	1.015 (0.848, 1.215)	1.008 (0.808, 1.258)	1.075 (0.901, 1.282)
Time of last meal						
Crude	−0.205 (−0.321, −0.089) *	−0.270 (−0.399, −0.142) *	1.184 (0.989, 1.416)	1.047 (0.879, 1.248)	1.331 (1.066, 1.661) *	1.354 (1.131, 1.622) *
Adjusted	−0.185 (−0.302, −0.068) *	−0.259 (−0.391, −0.128) *	1.150 (0.954, 1.386)	1.013 (0.84,3 1.216)	1.337 (1.048, 1.706) *	1.388 (1.141, 1.690) *
Time of largest meal						
Crude	−0.060 (−0.114, −0.007) *	−0.066 (−0.126, −0.007) *	1.110 (1.020, 1.207) *	1.096 (1.008, 1.192) *	1.083 (0.979, 1.199)	1.057 (0.978, 1.142)
Adjusted	−0.053 (−0.107, 0.002)	−0.069 (−0.130, −0.007) *	1.126 (1.028, 1.233) *	1.116 (1.019, 1.223) *	1.076 (0.960, 1.206)	1.058 (0.973, 1.150)
Eating window						
Crude	−0.316 (−0.404, −0.229) *	−0.114 (−0.217, −0.010) *	0.936 (0.814, 1.076)	0.891 (0.773, 1.027)	1.139 (0.965, 1.343)	1.068 (0.937, 1.218)
Adjusted	−0.292 (−0.383, −0.201) *	−0.088 (−0.196, 0.020)	0.941 (0.809, 1.094)	0.892 (0.765, 1.040)	1.197 (0.995, 1.439)	1.109 (0.960, 1.280)
Duration from last meal to bedtime						
Crude	−0.151 (−0.268, −0.034) *	−0.272 (−0.401, −0.143) *	0.729 (0.59,8 0.889) *	0.762 (0.627, 0.927) *	0.821 (0.651, 1.035)	0.957 (0.808, 1.132)
Adjusted	−0.176 (−0.293, −0.059) *	−0.283 (−0.414, −0.153) *	0.738 (0.601, 0.906) *	0.765 (0.626, 0.937) *	0.761 (0.591, 0.980) *	0.955 (0.795, 1.147)

^1^ Adjusted for age, education, socioeconomic status, marital status, smoking, alcohol, physical activity, mental health issues, and food insecurity. ^2^ Dichotomous outcomes indicating sleep latency > 30 min (1) or sleep latency ≤ 30 min (0). ^3^ Dichotomous outcome indicating poor sleep quality (rated as “fairly bad” or “very bad”, 1) or good sleep quality (rated as “fairly good” or “very good”, 0). * *p* < 0.05.

## Data Availability

The data presented in this study are available on request from the corresponding author due to privacy restrictions.

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
