# Peer review of "Meal Timing and Sleep Health Among Midlife Mexican Women During the Early Stages of the COVID-19 Pandemic"

_nutrients, 2024, doi:10.3390/nu16223967_

Round 1

Reviewer 1 Report

Comments and Suggestions for Authors

First of all, I would like to express my deep appreciation and gratitude for the opportunity to review this important study, "Meal Timing Habits in Relation to Sleep Health Among Mid-life Mexican Women during the Covid-19 Pandemic." It is a significant contribution that highlights crucial aspects of the relationship between meal timing and sleep quality, particularly in the under-researched context of midlife women during the pandemic. The clarity of the writing and the rigorous methodology make this work a valuable addition to the existing literature.

I would like to highlight some strengths of the work:

  • The study objective (lines 15-31) is clear and well-defined: investigating the relationship between eating habits and sleep health in midlife Mexican women is highly relevant, especially in the context of the pandemic.
  • The methodology used (lines 80-146) is solid and appropriate for the observational study design. The use of linear and logistic regressions for statistical analysis is appropriate and well justified.
  • The analysis of the results (lines 147-196) provides interesting insights, with significant associations emerging clearly, such as the inverse relationship between the eating window and weekday sleep duration.

However, there are a few areas that could be improved or clarified to enhance the manuscript’s message and comprehensibility:

  1. Title (line 2): The title could be more concise. I suggest modifying it to: "Meal Timing and Sleep Health in Midlife Mexican Women During the Covid-19 Pandemic" for better clarity and brevity.

  2. Abstract (lines 15-33): The abstract is well structured but could be improved by reducing repetitions and enhancing conciseness. I suggest condensing the results section to avoid overlapping with the "Results" section.

  3. Introduction (lines 35-79):

    • Lines 53-57: The phrase "Meal timing, including the clock time of individual meals..." could be made clearer by reformulating the concept of "daily eating window variability." I suggest: "Temporal variations in meal timing, including daily fluctuations in the eating window."
    • Lines 72-76: Additional details on the differences between peri-menopausal and post-menopausal women could be added, as they are only briefly mentioned but could be further explored to support the importance of the chosen sample.
  4. Results (lines 147-196):

    • Lines 186-189: I suggest rephrasing the concept of "meal timing was associated with sleep latency" to better specify the association with prolonged latency on both weekdays and weekends.
    • Table 3 (line 198): There is no clear indication of non-significant values. It would be helpful to better highlight when associations are not statistically significant to avoid misinterpretations.
  5. Discussion (lines 201-308):

    • Lines 214-217: It may be useful to add a more detailed comparison with previous studies that have addressed the association between meal timing and sleep quality.
    • Lines 235-237: I suggest eliminating this sentence, as the association between evening chronotype and sleep quality is not supported by the presented data and could confuse readers.
  6. Study limitations (lines 299-307): The authors briefly mention some limitations, such as the cross-sectional design and lack of data on food content. I suggest including:

    • The absence of objective sleep assessment via actigraphy, which would have been useful to confirm the self-reported results.
    • The potential recall bias, as the data were collected via phone survey.

I would kindly suggest including the following reference in your manuscript, as it may enhance the depth of your discussion on the role of self-regulation in eating behaviors, which is particularly relevant when analyzing meal timing and its effects on sleep health:

Diotaiuti, P., Girelli, L., Mancone, S., Valente, G., Bellizzi, F., Misiti, F., & Cavicchiolo, E. (2022). Psychometric properties and measurement invariance across gender of the Italian version of the tempest self-regulation questionnaire for eating adapted for young adults. Frontiers in psychology, 13, 941784. https://doi.org/10.3389/fpsyg.2022.941784

This article explores the psychometric properties of a self-regulation questionnaire specific to eating behaviors, which could provide a theoretical framework to better understand how self-regulatory processes in food consumption might influence meal timing and sleep quality, especially in relation to your findings on how later eating times can affect sleep latency and quality.

I recommend incorporating this citation in the Introduction (lines 71-79), where you discuss the role of eating timing in women’s health. It can also be cited in the Discussion section (lines 275-278) to strengthen your analysis of behavioral aspects influencing eating patterns and their subsequent impact on sleep.

Author Response

First of all, I would like to express my deep appreciation and gratitude for the opportunity to review this important study, "Meal Timing Habits in Relation to Sleep Health Among Mid-life Mexican Women during the Covid-19 Pandemic." It is a significant contribution that highlights crucial aspects of the relationship between meal timing and sleep quality, particularly in the under-researched context of midlife women during the pandemic. The clarity of the writing and the rigorous methodology make this work a valuable addition to the existing literature.

I would like to highlight some strengths of the work:

  • The study objective (lines 15-31) is clear and well-defined: investigating the relationship between eating habits and sleep health in midlife Mexican women is highly relevant, especially in the context of the pandemic.
  • The methodology used (lines 80-146) is solid and appropriate for the observational study design. The use of linear and logistic regressions for statistical analysis is appropriate and well justified.
  • The analysis of the results (lines 147-196) provides interesting insights, with significant associations emerging clearly, such as the inverse relationship between the eating window and weekday sleep duration.

Response: Thanks for your review and your positive feedback.

However, there are a few areas that could be improved or clarified to enhance the manuscript’s message and comprehensibility:

  1. Title (line 2): The title could be more concise. I suggest modifying it to: "Meal Timing and Sleep Health in Midlife Mexican Women During the Covid-19 Pandemic" for better clarity and brevity.

Response: Thanks, we agree with this suggestion and have changed it.

  1. Abstract (lines 15-33): The abstract is well structured but could be improved by reducing repetitions and enhancing conciseness. I suggest condensing the results section to avoid overlapping with the "Results" section.

Response: Thanks, we have rewritten the abstract results in order to be more concise.

  1. Introduction (lines 35-79):
    • Lines 53-57: The phrase "Meal timing, including the clock time of individual meals..." could be made clearer by reformulating the concept of "daily eating window variability." I suggest: "Temporal variations in meal timing, including daily fluctuations in the eating window."

Response: Thanks for this suggested re-phrasing, which we have incorporated.

“Temporal variations in meal timing, including the clock time (and circadian time) of individual meals, the length of the eating window (hours from first to last eating occasion), and daily fluctuations in the eating window or individual meals, is now being recognized to have a substantial impact on health, particularly metabolic health [16–19].”

    • Lines 72-76: Additional details on the differences between peri-menopausal and post-menopausal women could be added, as they are only briefly mentioned but could be further explored to support the importance of the chosen sample.

Response: Thanks, this is a good point. We have added some text regarding the higher prevalence of sleep disruptions specifically during the menopausal transition.

“Perimenopause is also a time when many women experience sleep disruptions (e.g., insomnia and nighttime awakenings) due to the menopausal transition[22].”

  1. Results (lines 147-196):
    • Lines 186-189: I suggest rephrasing the concept of "meal timing was associated with sleep latency" to better specify the association with prolonged latency on both weekdays and weekends.

Response: Thank you. We have removed this phrase and instead directly described the associations with prolonged sleep latency on both weekdays and weekends.

“Finally, a later time for the largest meal was linked to shorter sleep duration on weekends (β = -0.069, 95% CI [-0.130, -0.007]), as well as prolonged sleep latency in both weekdays (OR = 1.126, 95% CI [1.028, 1.233]) and weekends (OR = 1.116, 95% CI [1.019, 1.223]).”

    • Table 3 (line 198): There is no clear indication of non-significant values. It would be helpful to better highlight when associations are not statistically significant to avoid misinterpretations.

Response: We have now indicated with asterisks the associations that are statistically significant at P<0.05, so that it is more clear which associations are significant vs non-significant.

  1. Discussion (lines 201-308):
    • Lines 214-217: It may be useful to add a more detailed comparison with previous studies that have addressed the association between meal timing and sleep quality.

Response: We have added references to support the statement (previously starting on Line 214), and then include more details of these studies in the following sentences.

“The finding that later timing of eating- including later timing of the first meal, midday meal, the closest meal/snack to bedtime, and the largest meal (not necessarily mutual exclusive) was related to poorer sleep health is consistent with some previous studies      [19,20,24,37,38].”

    • Lines 235-237: I suggest eliminating this sentence, as the association between evening chronotype and sleep quality is not supported by the presented data and could confuse readers.

Response: Thanks. We agree and have eliminated the sentence.

  1. Study limitations (lines 299-307): The authors briefly mention some limitations, such as the cross-sectional design and lack of data on food content. I suggest including:
    • The absence of objective sleep assessment via actigraphy, which would have been useful to confirm the self-reported results.
    • The potential recall bias, as the data were collected via phone survey.

Response: As suggested, we have added these limitations.

“Moreover, information was collected via phone survey, which can be subject to recall bias. Related to the survey format of the Covid-19 study, we did not have access to objective sleep assessments via actigraphy in order to validate self-reported sleep duration.”

I would kindly suggest including the following reference in your manuscript, as it may enhance the depth of your discussion on the role of self-regulation in eating behaviors, which is particularly relevant when analyzing meal timing and its effects on sleep health:

Diotaiuti, P., Girelli, L., Mancone, S., Valente, G., Bellizzi, F., Misiti, F., & Cavicchiolo, E. (2022). Psychometric properties and measurement invariance across gender of the Italian version of the tempest self-regulation questionnaire for eating adapted for young adults. Frontiers in psychology, 13, 941784. https://doi.org/10.3389/fpsyg.2022.941784

This article explores the psychometric properties of a self-regulation questionnaire specific to eating behaviors, which could provide a theoretical framework to better understand how self-regulatory processes in food consumption might influence meal timing and sleep quality, especially in relation to your findings on how later eating times can affect sleep latency and quality.

I recommend incorporating this citation in the Introduction (lines 71-79), where you discuss the role of eating timing in women’s health. It can also be cited in the Discussion section (lines 275-278) to strengthen your analysis of behavioral aspects influencing eating patterns and their subsequent impact on sleep.

Response: Thank you for pointing us to this paper. It is a very interesting and relevant point that self-regulation behaviors could be impacting meal times in these women. As suggested, we have added some text regarding this point.

“One eating behavior that may be relevant to meal timing is self-regulation. Specifically, women may be manipulating their diet for health reasons[56], including by altering meal timing. However, self-regulation of eating may not be independently related to sleep outcomes and therefore would cause residual confounding.”

Reviewer 2 Report

Comments and Suggestions for Authors

The authors examined the associations between meal timing habits and sleep duration, sleep latency, and sleep quality in midlife Mexican women during the Covid-19 pandemic. The present study found that 1) later timing of eating meals was related to poorer sleep health, 2) a longer eating window (number of hours between consumption of first and last meal or snack) was related to shorter sleep duration in weekday, 3) a shorter duration between last meal and bedtime was related to longer sleep duration but worse sleep quality.

The findings obtained by this study are of interest and the manuscript is well-written. However, several points need clarifying. These are given below.

Major comments:

1. Why the authors paid attention to the associations between meal and sleep DURING the COVID-19 PANDEMIC? Although the effects of the Covid-19 pandemic on eating habits and sleep were mentioned in the Discussion section (page 9, lines 281 to 291), the authors noted that “the pandemic, which is another important contextual factor to highlight”, which means that the Covid-19 pandemic was not a main factor for fluctuating eating habits or sleep. The authors should describe how the Covid-19 pandemic has negatively impacted people's eating and sleeping habits in the Introduction section.

2. In subsection 2.3. Sleep Outcomes in the Materials and Methods section, the authors noted that “Sleep latency was dichotomized to normal (≤30 minutes/night) or prolonged (>30 minutes/night)” (page 3, lines 110 to 111). What is the rationale for dichotomization with 30 minutes as the threshold? The authors need to mention the evidence with or without research papers.

3. Also, in subsection 2.3., why the authors asked the participant about their sleep quality both in the last month and the overall sleep quality (page 3, lines 111 to 112)? For what purpose were the two sleep qualities rated separately?

4. For the listed data in Table 2 in the Results section, the F-test was not appropriate statistical method due to that the F-test is used to reveal equal variances between two groups. If you really believe that this statistical analysis was appropriate, please provide the evidence for your opinion. Or re-analyze these data using the appropriate statistical method.

Minor comments:

1. in the page 5, lines 159 to 161, the bolded parts of the following sentences, “Women with higher socioeconomic status who were physically active…” were not appropriate, because the correlations between these variables were not examined.

2. in the Table 2, please indicate in the space provided what the bolded words mean.

3. in the Table 3, it was not specified which dummy variable 0 or 1 refers to. I guess that the dummy variable 1 corresponded to “Prolonged sleep latency” and “good sleep quality”. However, the authors should describe clearly the definition of dummy variables for the logistic regression analysis in the Materials and Methods section.

4. in the page 6, line 179, the word, “0.1.041” to “1.041”.

5. in the page 7, line 188, the letter, “β” to “OR”.

6. in the page 7, line 194, add the closing brackets after the word, “0.906]”.

7. in the page 8, line 207, add the words, “last meal” after the words, “late timing of eating”.

8. in the page 9, line 256, the abbreviation, “OSA” was first used.

9. in the page 9, line 265, add the period after the words, “evening[42]”.

I hope these comments will be helpful.

Author Response

The authors examined the associations between meal timing habits and sleep duration, sleep latency, and sleep quality in midlife Mexican women during the Covid-19 pandemic. The present study found that 1) later timing of eating meals was related to poorer sleep health, 2) a longer eating window (number of hours between consumption of first and last meal or snack) was related to shorter sleep duration in weekday, 3) a shorter duration between last meal and bedtime was related to longer sleep duration but worse sleep quality.

The findings obtained by this study are of interest and the manuscript is well-written. However, several points need clarifying. These are given below.

Major comments:

  1. Why the authors paid attention to the associations between meal and sleep DURING the COVID-19 PANDEMIC? Although the effects of the Covid-19 pandemic on eating habits and sleep were mentioned in the Discussion section (page 9, lines 281 to 291), the authors noted that “the pandemic, which is another important contextual factor to highlight”, which means that the Covid-19 pandemic was not a main factor for fluctuating eating habits or sleep. The authors should describe how the Covid-19 pandemic has negatively impacted people's eating and sleeping habits in the Introduction section.

 Response: This is a good point. We have now added text in the introduction to describe the pandemic’s effects on sleep and eating behaviors.

“This study was completed during the early stages of the Covid-19 pandemic, a period of time when sleep and dietary patterns changed in many populations across the globe[26]. In midlife women specifically, declines in sleep quality were noted whereas changes in diet quality were mixed[27–30]. Although the goal of the present study was not to evaluate the effect of the pandemic on eating and sleep in this population, it is important to acknowledge the specific context during which the meal timing and sleep quality in this Mexico City population of midlife women were assessed.”

  1. In subsection 2.3. Sleep Outcomes in the Materials and Methods section, the authors noted that “Sleep latency was dichotomized to normal (≤30 minutes/night) or prolonged (>30 minutes/night)” (page 3, lines 110 to 111). What is the rationale for dichotomization with 30 minutes as the threshold? The authors need to mention the evidence with or without research papers.

 Response: Thank you, we recognize that using a cutoff of 30 minutes is fairly arbitrary, and clinical insomnia thresholds in fact do not include a threshold for what is considered a difficult time falling asleep. However, we used this cutoff based on prior literature, which we have now cited.

“Sleep latency was dichotomized to normal (≤30 minutes/night, 1) or prolonged (>30 minutes/night, 0), in accordance with prior literature [33,34].”

  1. Also, in subsection 2.3., why the authors asked the participant about their sleep quality both in the last month and the overall sleep quality (page 3, lines 111 to 112)? For what purpose were the two sleep qualities rated separately?

Response: We thought that the sleep quality experienced in the last month (i.e., during the pandemic) may not be representative of the longer-term overall sleep quality, which is why we analyzed both questions. Moreover, there were some differences in the associations between meal timing with sleep quality overall vs sleep quality in the last month. We have added some text regarding this finding, and we further clarified that “overall sleep quality” means longer-term usual sleep quality.

Ideally, we would have followed the same approach for sleep duration and sleep onset latency, but we were also trying to keep the survey fairly short. We have added some additional text to the limitations regarding this point.

“Another example of a potential role of the pandemic on results was the fact that the association between time from last meal to bedtime with sleep quality was only apparent for “overall sleep quality” (more representative of usual sleep quality) and not “sleep quality in last month” (more subject to circumstances related to the pandemic).”

“Finally, the short survey format also did not allow us to fully evaluate changes in sleep due to the pandemic, although we did ask about sleep quality in the previous month compared to longer-term sleep health.”

  1. For the listed data in Table 2 in the Results section, the F-test was not appropriate statistical method due to that the F-test is used to reveal equal variances between two groups. If you really believe that this statistical analysis was appropriate, please provide the evidence for your opinion. Or re-analyze these data using the appropriate statistical method.

 Response: Thanks, this is a good point. We have re-analyzed using a non-parametric Kruskal Wallis test and updated the table. There were a few associations that lost statistical significance, and we adjusted the results accordingly.

Minor comments:

  1. in the page 5, lines 159 to 161, the bolded parts of the following sentences, “Women with higher socioeconomic status who werephysically active…” were not appropriate, because the correlations between these variables were not examined.

Response: Thanks for catching this. We have revised the sentence.

  1. in the Table 2, please indicate in the space provided what the bolded words mean.

Response: We have added a footnote to indicate what the bolded words mean (P<0.05).

  1. in the Table 3, it was not specified which dummy variable 0 or 1 refers to. I guess that the dummy variable 1 corresponded to “Prolonged sleep latency” and “good sleep quality”. However, the authors should describe clearly the definition of dummy variables for the logistic regression analysis in the Materials and Methods section.

Response: Thanks. We have included more details in both the footnote of the table and the Methods.

  1. in the page 6, line 179, the word, “0.1.041” to “1.041”.

Response: It has been fixed.

  1. in the page 7, line 188, the letter, “β” to “OR”.

Response: It has been fixed.

  1. in the page 7, line 194, add the closing brackets after the word, “0.906]”.

Response: It has been fixed.

  1. in the page 8, line 207, add the words, “last meal” after the words, “late timing of eating”.

Response: It has been fixed.

  1. in the page 9, line 256, the abbreviation, “OSA” was first used.

Response: Thanks. We have spelled it out instead.

  1. in the page 9, line 265, add the period after the words, “evening[42]”.

Response: It has been fixed. Thanks for your attention to detail.

Round 2

Reviewer 1 Report

Comments and Suggestions for Authors

Kindest,

the work has been well reviewed. For me you can proceed to publication.

I wish the best

Author Response

Kindest,

the work has been well reviewed. For me you can proceed to publication.

I wish the best

Response: Thanks again for your review of our work!

Reviewer 2 Report

Comments and Suggestions for Authors

The authors clearly responded to almost all of my comments and suggestions. However, several concerns remain.

1) the authors added the following text in the introduction to describe the pandemic’s effects on sleep and eating behaviors: “This study was completed during the early stages of the Covid-19 pandemic, a period of time when sleep and dietary patterns changed in many populations across the globe[26].” I think that this limited focus of evaluation period in this study, “during the early stages of the Covid-19 pandemic” should be noted in the “Title” and “Abstract” sections.

2) the authors re-analyzed the data in the Table 2 using a non-parametric Kruskal Wallis test. However, the Kruskal-Wallis test is a statistical method for comparing data from multiple groups. Therefore, the Kruskal-Wallis test may be inappropriate for the statistical analysis for the two categorical variables that composed of two groups, “Physical activity” and “Mental health issues”. And the authors should clearly describe the statistical method in the subsection 2.5 Statistical Analysis in the “Materials and Methods” section.

3) Although the authors corrected the text, “Women with higher socioeconomic status who were physically active…”  to “Women with higher socioeconomic status, those who were physically active…”, it seems not appropriate. Because this study did not examine the relations among the attributes other than meal timing characteristics (e.g.,  such as age, education, marital status, and so on), the author should know whether women with higher socioeconomic status were actually physically active, and so on. It should be kept in mind that the relationship between meal timing characteristics and each of demographics and lifestyle characteristics was evaluated independently.

4) For the logistic regression analysis, the authors added the definition of dummy variables in the footnote of the Table 3. However, in the subsection 2.3 Sleep Outcomes in the “Materials and methods” section, the definition of dummy variables was still not sufficiently described. To point out specifically, in the page 3, line 118, the authors noted “normal (30 minutes/night, 1) or prolonged (>30 minutes/night, 0)”, however, the presentation of dummy variables was unfamiliar to the reader. Similarly, the following sentence in the page 3, lines 121 to 123, “We categorized sleep quality as poor (1) if the response was … and as good (0) if the response was… “ was uninformative. I strongly suggest that the definition of dummy variables for the logistic regression analysis is described in the subsection 2.5 Statistical Analysis in the “Materials and methods” section.

I hope these comments will be helpful.

Author Response

1) the authors added the following text in the introduction to describe the pandemic’s effects on sleep and eating behaviors: “This study was completed during the early stages of the Covid-19 pandemic, a period of time when sleep and dietary patterns changed in many populations across the globe[26].” I think that this limited focus of evaluation period in this study, “during the early stages of the Covid-19 pandemic” should be noted in the “Title” and “Abstract” sections.

Response: Thank you. We have updated as suggested. 

“Meal Timing and Sleep Health Among Midlife Mexican Women during the early stages of the Covid-19 Pandemic”

2) the authors re-analyzed the data in the Table 2 using a non-parametric Kruskal Wallis test. However, the Kruskal-Wallis test is a statistical method for comparing data from multiple groups. Therefore, the Kruskal-Wallis test may be inappropriate for the statistical analysis for the two categorical variables that composed of two groups, “Physical activity” and “Mental health issues”. And the authors should clearly describe the statistical method in the subsection 2.5 Statistical Analysis in the “Materials and Methods” section.

Response: Thanks, you are right that for two groups this test is actually a Mann Whitney U test instead of Kruskal Wallis. We have updated subsection 2.5 and the table footnote. 

3) Although the authors corrected the text, “Women with higher socioeconomic status who were physically active…”  to “Women with higher socioeconomic status, those who were physically active…”, it seems not appropriate. Because this study did not examine the relations among the attributes other than meal timing characteristics (e.g.,  such as age, education, marital status, and so on), the author should know whether women with higher socioeconomic status were actually physically active, and so on. It should be kept in mind that the relationship between meal timing characteristics and each of demographics and lifestyle characteristics was evaluated independently.

Response: We have re-phrased this sentence so that it’s hopefully now clear that the associations were each conducted independently.

“Higher socioeconomic status, being physically active, no alcohol consumption, and no food insecurity were each associated with a higher number of  meals and snacks consumed per day.” 

4) For the logistic regression analysis, the authors added the definition of dummy variables in the footnote of the Table 3. However, in the subsection 2.3 Sleep Outcomes in the “Materials and methods” section, the definition of dummy variables was still not sufficiently described. To point out specifically, in the page 3, line 118, the authors noted “normal (≤30 minutes/night, 1) or prolonged (>30 minutes/night, 0)”, however, the presentation of dummy variables was unfamiliar to the reader. Similarly, the following sentence in the page 3, lines 121 to 123, “We categorized sleep quality as poor (1) if the response was … and as good (0) if the response was… “ was uninformative. I strongly suggest that the definition of dummy variables for the logistic regression analysis is described in the subsection 2.5 Statistical Analysis in the “Materials and methods” section.

Response: As suggested, we have moved the description of the binary variables to the Statistical Analysis section. We hope that it is clearer now. 

“Logistic regression was used to examine the associations between study exposures and the binary outcomes prolonged sleep latency and poor sleep quality (in separate models). For each sleep outcome, 0 was the more optimal sleep category (sleep latency≤30 minutes or good sleep quality, rated as “fairly good” or “very good) and 1 was the less optimal (>30 minutes sleep latency or poor sleep quality, rated as “very bad” or “fairly bad”).”

I hope these comments will be helpful.

Response: Thanks again for your review, and we hope these further points have now been addressed.